# Synthesis and evaluation of radiogallium-labeled long-chain fatty acid derivatives as myocardial metabolic imaging agents

**Nurmaya Effendi[1,2], Kenji Mishiro[1], Hiroshi Wakabayashi[3], Malwina Gabryel-Skrodzka[4], Kazuhiro Shiba[5], Junichi Taki[3], Renata Jastrząb[4], Seigo Kinuya[3], Kazuma Ogawa[1,6]***

**1** Institute for Frontier Science Initiative, Kanazawa University, Kakuma-machi, Kanazawa, Ishikawa, Japan, **2** Faculty of Pharmacy, Universitas Muslim Indonesia, Makassar, South Sulawesi, Indonesia, **3** Department of Nuclear Medicine, Kanazawa University Hospital, Kanazawa University, Takara-machi, Kanazawa, Ishikawa, Japan, **4** Faculty of Chemistry, Adam Mickiewicz University in Poznan, Poznan, Poland, **5** Research Center for Experimental Modeling of Human Disease, Kanazawa University, Takara-machi, Kanazawa, Ishikawa, Japan, **6** Graduate School of Medical Sciences, Kanazawa University, Kakuma-machi, Kanazawa, Ishikawa, Japan

* kogawa@p.kanazawa-u.ac.jp

**Data Availability Statement:** All relevant data are within the paper and its Supporting Information files.

## Abstract

Since long-chain fatty acids work as the primary energy source for the myocardium, radiolabeled long-chain fatty acids play an important role as imaging agents to diagnose metabolic heart dysfunction and heart diseases. With the aim of developing radiogallium-labeled fatty acids, herein four fatty acid-based tracers, [$^{67}$Ga]Ga-HBED-CC-PDA, [$^{67}$Ga]Ga-HBED-CC-MHDA, [$^{67}$Ga]Ga-DOTA-PDA, and [$^{67}$Ga]Ga-DOTA-MHDA, which are [$^{67}$Ga]Ga-HBED-CC and [$^{67}$Ga]Ga-DOTA conjugated with pentadecanoic acid (PDA) and 3-methylhexadecanoic acid (MHDA), were synthesized, and their potential for myocardial metabolic imaging was evaluated. Those tracers were found to be chemically stable in 0.1 M phosphate buffered saline. Initial [$^{67}$Ga]Ga-HBED-CC-PDA, [$^{67}$Ga]Ga-HBED-CC-MHDA, [$^{67}$Ga]Ga-DOTA-PDA, and [$^{67}$Ga]Ga-DOTA-MHDA uptakes in the heart at 0.5 min postinjection were 5.01 ± 0.30%ID/g, 5.74 ± 1.02%ID/g, 5.67 ± 0.22%ID/g, and 5.29 ± 0.10%ID/g, respectively. These values were significantly lower than that of [$^{123}$I]BMIPP (21.36 ± 2.73%ID/g). For their clinical application as myocardial metabolic imaging agents, further structural modifications are required to increase their uptake in the heart.

## Introduction

Long-chain fatty acids are the predominant energy substrate for the healthy myocardium and are metabolized by $\beta$-oxidation in the heart [1]. With myocardial abnormalities, such as ischemic disease and cardiomyopathy, the alteration of cardiac fuel metabolism is mostly observed. Herein glycolysis and glycogen metabolism become the primary energy source, whereas fatty acid oxidation is suppressed. Thus, alteration in fatty acid metabolism has been identified as a biomarker for ischemia and myocardial damage [2–4]. Hence, either single-photon emission computed tomography (SPECT) or positron emission tomography (PET) imaging with radiolabeled long-

**Funding:** This work was supported in part by Suzuken Memorial Foundation and Kanazawa University SAKIGAKE project 2020. The funders had no role in study design, data collection and analysis, decision to publish, or preparation of the manuscript.

**Competing interests:** The authors have declared that no competing interests exist.

chain fatty acid analogs as radiotracers has been considered as a valuable tool for the early detection of myocardial disease, such as unstable angina and severe heart ischemia [5].

With regard to SPECT tracers, one of the early radioiodinated fatty acids is 15-($p$-[$^{123}$I] iodophenyl)pentadecanoic acid ([$^{123}$I]IPPA, Fig 1), which exhibited fast heart uptake. However, its rapid washout from the heart has limited its clinical use. Alternatively, a radioiodinated fatty acid analog with methyl-branch, $\beta$-methyl [$^{123}$I]iodophenyl-pentadecanoic acid ([$^{123}$I]BMIPP, Fig 1), was developed, which is slowly washed out from the heart and shows better image quality than [$^{123}$I]IPPA [6,7]. Hence, [$^{123}$I]BMIPP is a more promising myocardial metabolic imaging agent, and has been approved for the diagnosis of heart dysfunction in Japan since 1993 [8].

Due to accurate attenuation correction, higher spatial resolution, and shorter image acquisition time, the ability to visualize myocardial conditions of PET is generally considered superior than that of SPECT. A PET tracer [1-$^{11}$C]palmitate (Fig 1) exhibited uptake and clearance patterns, which were correlated with fatty acid $\beta$-oxidation, in the heart [7]. However, the short half-life of $^{11}$C (20 min) limits its clinical use. Alternatively, radiofluorinated fatty acid analogs, such as [$^{18}$F]fluoro-6-thia-heptadecanoic acid ([$^{18}$F]FTHA, Fig 1) and trans-9($RS$)-[$^{18}$F]fluoro-3,4-($RS$,$RS$)methyleneheptadecanoic acid ([$^{18}$F]FCPHA, Fig 1), have been explored for the diagnosis of myocardial dysfunction [9,10]. However, rapid clearance by $\beta$-oxidation in the heart and slow washout from the intracellular lipid pool limited their application in clinical practice for the early detection of ischemic diseases. Development of radiogallium-labeled fatty acid derivatives is an alternative for PET tracers with on-site cyclotron-produced radionuclides, such as C-11 and F-18. $^{68}$Ga, one of the positron emitters, can be eluted on demand from a portable $^{68}$Ge/$^{68}$Ga generator system, obviating a cyclotron [11]. Previously, $^{68}$Ga-labeled fatty acids with 1,4,7-triazacyclonane-1,4,7-triacetic acid (NOTA), 1,4,7-triazacyclononane-1-glutaric acid-4,7-acetic acid (NODAGA), and diethylenetriaminepentaacetic acid (DTPA) as chelators were reported [12–14]. However, their low heart/blood ratios and high accumulation in non-targeted tissues, such as the liver, limited their potential as myocardial metabolic imaging agents.

Otto *et al.* found that an increase in the number of the carbon chain until 21 in fatty acids has been shown to improve the myocardial uptake [15]. Besides, another study verified that methyl-branched fatty acids can prolong cardiac retention [16]. As [$^{123}$I]BMIPP with 15 carbon chains is well established, we chose lead compounds with similar length of carbon chains, pentadecanoic acid (PDA) as an unbranched fatty acid and 3-methylhexadecanoic acid (MHDA) as a methyl-branched fatty acid, for the preparation of radiogallium labeled fatty acids for myocardial metabolic imaging.

In this study, we conjugated alkyl-unbranched and methyl-branched fatty acids with two types of chelating agents to form stable Ga complexes, which have distinctive lipophilicities. Here, we used $N$,$N'$-bis-[2-hydroxy-5-(carboxyethyl)benzyl]ethylenediamine-$N$,$N'$-diacetic acid (HBED-CC) as an acyclic chelating agent and $S$-2-(4-isothiocyanatobenzyl)-1,4,7,10,tetraazacyclododecane tetraacetic acid (DOTA-Bn-SCN) as a macrocyclic chelating agent. Then, four fatty acid-based tracers, [$^{67}$Ga]Ga-HBED-CC-PDA ([$^{67}$Ga]**5**), [$^{67}$Ga]Ga-HBED-CC-MHDA ([$^{67}$Ga]**6**), [$^{67}$Ga]Ga-DOTA-PDA ([$^{67}$Ga]**7**), and [$^{67}$Ga]Ga-DOTA-MHDA ([$^{67}$Ga]**8**) (Fig 1), with an easy-to-handle radioisotope $^{67}$Ga, which has a longer half-life (t$_{1/2}$ = 3.3 days) than $^{68}$Ga (t$_{1/2}$ = 68 min) were prepared and evaluated to assess their feasibilities as myocardial imaging agents.

## Materials and methods

### General

Commercial reagents and solvents were purchased from Sigma-Aldrich (St. Louis, MO, USA), Wako Pure Chemical Industries (Osaka, Japan), Nacalai Tesque, Inc., (Kyoto, Japan), Tokyo

**Fig 1. Chemical structures of [123I]BMIPP, [123I]IPPA, [1-11C]palmitate, [18F]FCPHA, [18F]FTHA, [67Ga] Ga-HBED-CC-PDA ([67Ga]5), [67Ga]Ga-HBED-CC-MHDA ([67Ga]6), [67Ga]Ga-DOTA-PDA ([67Ga]7), and [67Ga] Ga-DOTA-MHDA ([67Ga]8).**

Chemical Industry, Co., Ltd., (Tokyo, Japan), and Macrocyclics (Dallas, TX, USA). All of these were used without further purification unless otherwise stated.

[67Ga]GaCl$_3$ and [123I]BMIPP were supplied by Nihon Medi-Physics Co., Ltd. (Tokyo, Japan). The radioactivity was measured by an Auto Gamma System ARC-7010B (Hitachi, Ltd., Tokyo, Japan).

Proton nuclear magnetic resonance (1H-NMR) spectra was recorded on JEOL JNM-ECS400 (JEOL Ltd, Tokyo, Japan). Figures of the NMR spectra are available in the Supplementary Material file (S6 Fig). Electrospray ionization mass spectra (ESI-MS) were obtained with JEOL JMS-T100TD (JEOL Ltd). TLC analysis was performed using silica plates (Art 5553, Merck, Darmstadt, Germany).

## Synthesis of precursors and reference compounds

The detailed synthesis procedures of intermediate compounds, 15-aminopentadecanoic acid (S1 File), 16-amino-3-methylhexadecanoic acid (S2 File), and HBED-CC(tBu)$_3$-NHS ester (S3 File) were described in the Supporting Information.

**15-[3-(3-{[(2-{[5-(2-Carboxyethyl)-2-hydroxybenzyl](carboxymethyl)amino}ethyl)(carboxymethyl)amino]methyl}-4-hydroxyphenyl)propanamido]pentadecanoic acid (HBED-CC-PDA)** (1)

A mixture of 15-aminopentadecanoic acid (4.6 mg, 18.0 μmol), N, N-diisopropylethylamine (DIPEA) (15 μL, 81.0 μmol,), and HBED-CC(tBu)$_3$-NHS ester (16.8 mg, 20.0 μmol) in dry dichloroethane (1 mL) was stirred at 50˚C for 2 h under nitrogen atmosphere. After completion of the reaction, the reaction mixture was concentrated under reduced pressure. TFA was added to the residue and the mixture was stirred for 2 h. After removing TFA by nitrogen gassing, the crude product was purified by RP-HPLC using a Cosmosil 5C$_{18}$-AR-II column (10 mm ID × 250 mm; Nacalai Tesque) at flow rate of 4 mL/min with a gradient mobile phase of 76% methanol in water with 0.1% TFA to 82% methanol in water with 0.1% TFA for 10 min. The column temperature was 40˚C. The fraction containing **1** was determined by

ESI-MS and collected. After removing the solvent by freeze-dryer, **1** was obtained as a colorless solid (4.2 mg, 30%).

$^1$H NMR (400 MHz, DMSO-$d_6$): δ 7.75 (1H, t, $J$ = 5.8 Hz), 7.04–6.98 (4H, m), 6.76–6.73 (2H, m), 3.91 (4H, s), 3.53 (4H, s), 3.07–2.97 (6H, m), 2.71–2.66 (4H, m), 2.51–2.43 (2H, m), 2.30–2.26 (2H, m), 2.18 (2H, t, $J$ = 7.2 Hz), 1.49–1.11 (24H, m). LRMS (ESI+), calcd for $C_{41}H_{61}N_3O_{11}$ [M+H$^+$]: m/z = 772.4, found 772.3.

### 16-[3-(3-{[(2-{5-(2-Carboxyethyl)-2-hydroxybenzyl](carboxymethyl)amino} ethyl)(carboxymethyl)amino]methyl}-4-hydroxyphenyl)propanamido] -3-methylhexadecanoic acid (HBED-CC-MHDA) (2)

A mixture of 16-amino-3-methylhexadecanoic acid (14.3 mg, 50.0 μmol), DIPEA (38 μL, 0.2 mmol), and HBED-CC(tBu)$_3$-NHS ester (60 mg, 75 μmol) in dry dichloroethane (1 mL) was stirred at 50˚C for 2 h under nitrogen atmosphere. After completion of the reaction, the reaction mixture was concentrated under reduced pressure. TFA was added to the residue and the mixture was stirred for 2 h. After removing TFA by nitrogen gassing, the crude product was purified by RP-HPLC using a Cosmosil 5C$_{18}$-AR-II column (10 mm ID × 250 mm) at flow rate of 4 mL/min with a gradient mobile phase of 78% methanol in water with 0.1% TFA to 84% methanol in water with 0.1% TFA for 10 min. The column temperature was 40˚C. The fraction containing **2** was determined by ESI-MS and collected. After removing the solvent by freeze-dryer, **2** was obtained as a colorless solid (14 mg, 35%).

$^1$H NMR (400 MHz, DMSO-$d_6$): δ 7.77 (1H, t, $J$ = 5.2 Hz), 7.01–6.96 (4H, m), 6.74–6.70 (2H, m), 3.84 (4H, s), 3.44 (4H, s), 2.98–2.96 (6H, m), 2.71–2.65 (4H, m), 2.47–2.43 (2H, m), 2.30–2.25 (2H, m), 2.21–2.16 (1H, m), 2.01–1.96 (1H, m), 1.79 (1H, brs), 1.34–1.23 (24H, m), 0.87 (3H, d, $J$ = 6.8 Hz). LRMS (ESI+), calcd. for $C_{43}H_{65}N_3O_{11}$ [M+H$^+$]: m/z = 800.4, found 800.1.

### 2,2′,2″,2‴-(2-{4-[3-(14-Carboxytetradecyl)thioureido]benzyl}-1,4,7, 10-tetraazacyclododecane-1,4,7,10-tetrayl)tetraacetic acid (DOTA-PDA) (3)

A mixture of 15-aminopentadecanoic acid (1.1 mg, 4.2 μmol, 1.0 eq.), DOTA-Bn-SCN (3.5 mg, 5.1 μmol, 1.2 eq.), and DIPEA (21.2 μL, 127.2 μmol, 30.0 eq.) in DMF (0.1 mL) was stirred at rt for overnight. Upon completion of reaction, DMF was removed by nitrogen gassing. The crude product was purified by RP-HPLC using a Cosmosil 5C$_{18}$-AR-II column (10 mm ID × 250 mm) at flow rate of 4 mL/min with a gradient mobile phase of 72% methanol in water with 0.1% TFA to 85% methanol in water with 0.1% TFA for 10 min. The column temperature was 40˚C. The fraction containing **3** was determined by ESI-MS and collected. After removing the solvent by freeze-dryer, **3** was obtained as a colorless solid (2.8 mg, 82%).

$^1$H NMR (400 MHz, CD$_3$OD): δ 7.38 (2H, d, $J$ = 6.8 Hz), 7.29 (2H, d, $J$ = 8.0 Hz), 4.42–2.55 (27H, m), 2.28 (2H, t, $J$ = 6.4 Hz), 1.64–1.53 (4H, m), 1.35–1.30 (20H, m). LRMS (ESI+), calcd for $C_{39}H_{64}N_6O_{10}S$ [M+H$^+$]: m/z = 809.4, found 809.2.

### 2,2′,2″,2‴-(2-{4-[3-(15-Carboxy-14-methylpentadecyl)thioureido]benzyl}-1,4,7, 10-tetraazacyclododecane-1,4,7,10-tetrayl)tetraacetic acid (DOTA-MHDA) (4)

A mixture of 16-amino-3-methylhexadecanoic acid (1.2 mg, 4.2 μmol), DOTA-Bn-SCN (3.5 mg, 5.1 μmol), and DIPEA (21.2 μL, 127.2 μmol) in DMF (0.1 mL) was stirred at rt for overnight. After completion of the reaction, the reaction mixture was concentrated. The crude product was purified by RP-HPLC using a Cosmosil 5C$_{18}$-AR-II column (10 mm ID × 250 mm) at flow rate of 4 mL/min with a gradient mobile phase of 75% methanol in water with

0.1% TFA to 88% methanol in water with 0.1% TFA for 10 min. The column temperature was 40˚C. The fraction containing **4** was determined by ESI-MS and collected. After removing the solvent by freeze-dryer, **4** was obtained as a colorless solid (2.8 mg, 80%).

$^1$H NMR (400 MHz, CD$_3$OD): δ 7.37 (2H, d, $J$ = 8.4 Hz), 7.28 (2H, d, $J$ = 8.4 Hz), 4.31–2.56 (27H, m), 2.30–2.25 (1H, m), 2.10–2.04 (1H, m), 1.91 (1H, s-br), 1.58–1.63 (2H, m), 1.36–1.21 (22H, m), 0.94 (3H, d, $J$ = 6.8 Hz). LRMS (ESI+), calcd for C$_{41}$H$_{68}$N$_6$O$_{10}$S [M+H$^+$]: m/z = 837.4, found 837.2.

> **{15-[3-(3-{[(2-{[5-(2-Carboxyethyl)-2-hydroxybenzyl](carboxymethyl)amino}
> ethyl)(carboxymethyl)amino]methyl}-4-hydroxyphenyl)propanamido]
> pentadecanoic acid}gallium(III) (Ga-HBED-CC-PDA)** (5)

To a solution of **1** (1.0 mg, 1.3 μmol) in the mixed solvent of methanol and water (1/1) (100 μL), was added a solution of Ga(NO$_3$)$_3$ (9.9 mg, 39.0 μmol) in water (50 μL). The mixture was reacted at 40˚C for 4 h. The crude product was purified by RP-HPLC using a Cosmosil 5C$_{18}$-AR-II column (10 mm ID × 250 mm) at flow rate of 4 mL/min with a gradient mobile phase of 78% methanol in water with 0.1% TFA to 88% methanol in water with 0.1% TFA for 10 min. The column temperature was 40˚C. The fraction containing **5** was determined by ESI-MS and collected. After removing the solvent by freeze-dryer, **5** was obtained as a colorless solid (0.8 mg, 73%).

LRMS (ESI+), calcd for C$_{41}$H$_{57}$GaN$_3$O$_{11}$ [M+H$^+$]: m/z = 837.3, found 838.1.

> **{16-[3-(3-{[(2-{[5-(2-Carboxyethyl)-2-hydroxybenzyl](carboxymethyl)amino}
> ethyl)(carboxymethyl)amino]methyl}-4-hydroxyphenyl)propanamido]
> 3-methylhexadecanoic acid}gallium(III) (Ga-HBED-CC-MHDA)** (6)

To a solution of **2** (3.0 mg, 3.8 μmol) in mixed solvent of methanol and water (1/1) (200 μL), was added a solution of Ga(NO$_3$)$_3$ (28.7 mg, 112.5 μmol) in water (100 μL). The mixture was reacted at 40˚C for 4 h. The crude product was purified by RP-HPLC using a Cosmosil 5C$_{18}$-AR-II column (10 mm ID × 250 mm) at flow rate of 4 mL/min with a gradient mobile phase of 78% methanol in water with 0.1% TFA to 84% methanol in water with 0.1% TFA for 10 min. The column temperature was 40˚C. The fraction containing **6** was determined by ESI-MS and collected. After removing the solvent by freeze-dryer, **6** was obtained as a colorless solid (2.0 mg, 77%).

LRMS (ESI+), calcd for C$_{43}$H$_{61}$GaN$_3$O$_{11}$ [M+H$^+$]: m/z = 865.4, found 865.9.

> **[2,2′,2″,2‴-(2-{4-[3-(14-Carboxytetradecyl)thioureido]benzyl}-1,4,7,10-tetraazacy
> clododecane-1,4,7,10-tetrayl)tetraacetic acid]gallium(III) (Ga-DOTA-PDA)** (7)

To a solution of **3** (0.8 mg, 1.0 μmol) in mixed solvent of methanol and water (1/1) (100 μL), was added a solution of Ga(NO$_3$)$_3$ (7.6 mg, 30.0 μmole) in water (50 μL). The mixture was reacted at 40˚C for 4 h. The crude product was purified by RP-HPLC using a Cosmosil 5C$_{18}$-AR-II column (10 mm ID × 250 mm) at flow rate of 4 mL/min with a gradient mobile phase of 72% methanol in water with 0.1% TFA to 85% methanol in water with 0.1% TFA for 10 min. The column temperature was 40˚C. The fraction containing **7** was determined by ESI-MS and collected. After removing the solvent by freeze-dryer, **7** was obtained as a colorless solid (0.5 mg, 50%).

LRMS (ESI+), calcd for $C_{39}H_{62}GaN_6O_{10}S$ [M+H$^+$]: m/z = 876.4, found 876.3.

**[2,2′,2″,2‴-(2-{4-[3-(15-Carboxy-14-methylpentadecyl)thioureido]benzyl}-1,4,7,10-tetraaza cyclododecane-1,4,7,10-tetrayl)tetraacetic acid]gallium(III) (Ga-DOTA-MHDA)** (8)

To a solution of **4** (2.0 mg, 2.4 μmol) in mixed solvent of methanol and water (1/1) (200 μL), was added a solution of Ga(NO$_3$)$_3$ (18.2 mg, 72 μmol) in water (100 μL). The mixture was reacted at 40˚C for 4 h. The crude product was purified by RP-HPLC using a Cosmosil 5C$_{18}$-AR-II column (10 mm ID × 250 mm) at flow rate of 4 mL/min with a gradient mobile phase of 75% methanol in water with 0.1% TFA to 88% methanol in water with 0.1% TFA for 10 min. The column temperature was 40˚C. The fraction containing **8** was determined by ESI-MS and collected. After removing the solvent by freeze-dryer, **8** was obtained as a colorless solid (1.0 mg, 48%).

LRMS (ESI+), calcd for $C_{41}H_{66}GaN_6O_{10}S$ [M+H$^+$]: m/z = 904.4, found 904.6.

## Radiolabeling

Radiotracers, [$^{67}$Ga]Ga-HBED-CC-PDA ([$^{67}$Ga]**5**), [$^{67}$Ga]Ga-HBED-CC-MHDA ([$^{67}$Ga]**6**), [$^{67}$Ga]Ga-DOTA-PDA ([$^{67}$Ga]**7**), and [$^{67}$Ga]Ga-DOTA-MHDA ([$^{67}$Ga]**8**) were prepared by reaction between [$^{67}$Ga]GaCl$_3$ with 20 μg of precursors (**1**, **2**, **3**, and **4**, respectively) in 1.0 M HEPES buffer or 0.2 M ammonium acetate buffer pH 5.0 at 85˚C for 10 min. The radiotracers, [$^{67}$Ga]**5**, [$^{67}$Ga]**6**, [$^{67}$Ga]**7**, and [$^{67}$Ga]**8** were isolated by RP-HPLC using a Cosmosil 5C$_{18}$AR-II column (4.6 mm ID × 150 mm; Nacalai Tesque) at the flow rate of 1 mL/min with a gradient mobile phase of 70% methanol in water with 0.1% TFA to 95% methanol in water with 0.1% TFA for 20 min. The column temperature was maintained at 40˚C. Radiochemical yield and purity were determined by an auto well gamma counter.

## Determination of partition coefficients

Partition coefficients of [$^{67}$Ga]**5**, [$^{67}$Ga]**6**, [$^{67}$Ga]**7**, and [$^{67}$Ga]**8** into n-octanol and 0.1 M phosphate buffered saline (PBS) pH 7.4 were determined according to previously described procedures [17]. The partition coefficient was calculated by the ratio of cpm/mL in n-octanol to that in PBS, and expressed as a common logarithm (log P).

## In vitro stability assay in buffer and plasma

The stabilities of radiotracers, [$^{67}$Ga]**5**, [$^{67}$Ga]**6**, [$^{67}$Ga]**7**, and [$^{67}$Ga]**8**, in buffer and murine plasma, were analyzed according to previous procedures [18].

## In vitro protein-binding assays

The protein-binding of radiotracers, [$^{67}$Ga]**5**, [$^{67}$Ga]**6**, [$^{67}$Ga]**7**, and [$^{67}$Ga]**8**, in HSA were performed using ultrafiltration (vivacon® 500, Sartorius, Goettingen, Germany). In vitro protein-binding assays were carried out according to previous procedures [19].

## Animal study

The animal experimental protocols used were approved by the Committee on Animal Experimentation of Kanazawa University (AP-204165). All experiments with animals were conducted in strict accordance with the Guidelines for the Care and Use of Laboratory Animals of Kanazawa University. Mice were housed together in individually ventilated cages with three mice per cage. The animals were housed at 23˚C with a 12 h light/dark schedule with free access to a standardized mouse diet and provided drinking water *ad libitum* water. Autoclaved wood chips

were applied as bedding material. The behavior of the mice during the experimental period was normal. Seventy-five ddY mice (male; 6-week-old; 29–33 g) were purchased from Japan SLC Inc., Hamamatsu, Japan. The mice were randomly divided into five groups (n = 15 mice per group) with five different time points for each group (n = 3 mice per time point).

### Biodistribution study

Radiotracers, [$^{67}$Ga]**5**, [$^{67}$Ga]**6**, [$^{67}$Ga]**7**, or [$^{67}$Ga]**8**, in saline containing 5% ethanol and 0.1% tween-80, or [$^{123}$I]BMIPP (74 kBq/100 μL, respectively) was intravenously injected into mice via the tail. The ddY mice were sacrificed by decapitation at 0.5, 2, 5, 20, and 60 min. Tissues of interest were removed and weighed. The radioactivity of the tissues was determined using an auto well gamma counter. The data were expressed as percent injected dose per gram tissue (% ID/g).

### Metabolite analysis

According to the previous procedure, the metabolite analysis of radiotracers was carried out with slight modification [20]. Liver, heart, and blood were collected at 10 min postinjection of [$^{67}$Ga]**5**, [$^{67}$Ga]**6**, [$^{67}$Ga]**7**, or [$^{67}$Ga]**8** (1.85 MBq/100 μL,) into ddY mice. The blood was centrifuged at 2,000*g* for 10 min at 4°C. An equal volume of cold acetonitrile was added to the plasma, and liver and heart were homogenized in a 1:1 mixture of acetonitrile and water (1 g organ/10 mL mixed solvent) followed by centrifugation at 2,000*g* for 10 min at 4°C. All supernatants (plasma, liver, and heart) were analyzed by TLC and RP-HPLC.

## Results and discussion

### Synthesis of reference compounds and precursors

Long-chain fatty acid derivatives, unbranched fatty acid (15-aminopentadecanoic acid, APDA) and methyl-branched fatty acid (16-amino-3-methylhexadecanoic acid, AMHDA), were readily synthesized according to detailed procedures in Supporting Information. Precursors **1** and **2** were prepared by incorporating 15-aminopentadecanoic acid or 16-amino-3-methylhexadecanoic acid with synthesized chelator HBED-CC-tris(*t*Bu)-NHS ester, followed by removing the protecting group using TFA (Fig 2). Meanwhile, precursors **3** and **4** were synthesized by conjugating 15-aminopentadecanoic acid or 16-amino-3-methylhexadecanoic acid with DOTA-Bn-SCN as a chelating agent (Fig 3). Thus, the precursors were prepared with high purities. The high purities of the precursors were confirmed by HPLC. The chromatograms of the precursors are shown in S1 Fig. Subsequently, non-radioactive complexes (**5**, **6**, **7**, and **8**) were synthesized with yields of 73%, 77%, 50%, and 48%, respectively. These non-radioactive gallium complexes were used as reference compounds for radiolabeled fatty acid derivatives.

### Synthesis of radiolabeled compounds

Four novel radiogallium-labeled fatty acid derivatives ([$^{67}$Ga]**5**, [$^{67}$Ga]**6**, [$^{67}$Ga]**7**, and [$^{67}$Ga]**8**) were prepared. The comparable retention times in the chromatograms (S2–S5 Figs) indicate that the radiolabeled products ([$^{67}$Ga]**5**, [$^{67}$Ga]**6**, [$^{67}$Ga]**7**, and [$^{67}$Ga]**8**) were identical to their nonradioactive counterparts (**5**, **6**, **7**, and **8**), determined using mass spectrometry. [$^{67}$Ga]**5** and [$^{67}$Ga]**6** with a HBED ligand were prepared with high radiochemical yields (>90%) and high radiochemical purities (>99%), whereas [$^{67}$Ga]**7** and [$^{67}$Ga]**8** with a DOTA ligand were prepared with moderate radiochemical yields, 33% and 36%, respectively, and high radiochemical purity (>98%) (Table 1). The radiolabeling was performed with carrier-free

**Fig 2. Synthesis of HBED-CC-PDA (1) and HBED-CC-MHDA (2).** i) DIPEA, dichloroethane, 50°C, 2 h; ii) TFA, rt, 2 h.

radionuclides and under non-carrier-added conditions. [$^{67}$Ga]5 and [$^{67}$Ga]6 were entirely separated from the corresponding precursors, 1 and 2, by the HPLC purification. Thus, the molar activity of [$^{67}$Ga]5 and [$^{67}$Ga]6 was estimated to be approximately $1.5 \times 10^{18}$ Bq/mol. Meanwhile, the molar activity of [$^{67}$Ga]7 and [$^{67}$Ga]8 was determined to be $1.1 \times 10^{15}$ Bq/mol and $1.2 \times 10^{15}$ Bq/mol, respectively. These values were much lower than that of [$^{67}$Ga]5 and [$^{67}$Ga]6 because [$^{67}$Ga]7 and [$^{67}$Ga]8 were not entirely separated from the corresponding precursors.

Previously, HBED-CC and DOTA have been reportedly used as a chelate site with different active molecules. For example, Eder *et al.* prepared a [$^{68}$Ga]Ga-HBED-CC complex conjugated with a prostate specific membrane antigen (PSMA) ligand, with a radiochemical yield and purity of >99% [21]. This complex is known as PSMA-11 and it has been approved as a PET radiopharmaceutical for prostate cancer imaging by European Medicines Agency (EMA) in 2019 and Food and Drug Administration (FDA) in 2020 [22]. High radiochemical yields (~ 95%) of [$^{68}$Ga]Ga-DOTA conjugated with octreotide, [$^{68}$Ga]Ga-DOTA-TOC, were also

**Fig 3. Synthesis of DOTA-PDA (3) and DOTA-MHDA (4).** i) DIPEA, DMF, rt, overnight.

**Table 1. Physical properties for radiogallium complex conjugated fatty acid derivatives.**

| Radiotracers | Physical properties | | | |
|---|---|---|---|---|
| | RC yield | RC purity | $t_R$ (min) | Log $P$ |
| [67Ga]Ga-HBED-CC-PDA ([67Ga]**5**) | 96.9% | 99.5% | 12.95 | $-1.64 \pm 0.00$ |
| [67Ga]Ga-HBED-CC-MHDA ([67Ga]**6**) | 94.5% | 99.3% | 16.16 | $-1.33 \pm 0.02$ |
| [67Ga]Ga-DOTA-PDA ([67Ga]**7**) | 33.9% | 99.0% | 10.36 | $-2.07 \pm 0.02$ |
| [67Ga]Ga-DOTA-MHDA ([67Ga]**8**) | 36.3% | 98.7% | 13.30 | $-1.39 \pm 0.01$ |

$t_R$ means retention time.

RC means radiochemical.

reported [23]. [68Ga]Ga-DOTA-TOC has been approved as a PET radiopharmaceutical agent by EMA in 2016 and FDA in 2019 for SSTR-positive neuroendocrine tumor imaging [24].

Previous basic researches demonstrated that HBED-CC conjugated bioactive molecules can be labeled with radiogallium in high radiochemical yields in the majority of papers. HBED-CC was proposed as an efficient radiogallium chelator with fast complexing kinetics and high stability [25,26]. Meanwhile, some DOTA-conjugated bioactive molecules can be labeled with radiogallium in high radiochemical yields; however, some DOTA-conjugated bioactive molecules can be labeled by radiogallium in moderate radiochemical yields [27–30]. Some bioactive molecules may interfere with the high yield of Ga-DOTA complexation. Different radiochemical yields between Ga-DOTA and Ga-HBED-CC complexes in this study and in previous studies may be derived from the difference of stability constants (log $K$), i.e., 21.3 and 38.5 respectively [31].

## Lipophilicity of 67Ga-labeled fatty acid derivatives

Not only log $P$-values of compounds but also their retention times using same HPLC conditions must be considered as indexes for lipophilicity. The order of lipophilicity of the four 67Ga-labeled fatty acid derivatives based on their log $P$-values and retention times was similar: [67Ga]**6** > [67Ga]**8** > [67Ga]**5** > [67Ga]**7** (Table 1).

As expected, the lipophilicity of radiogallium-labeled methyl-branched fatty acid ([67Ga]**6** and [67Ga]**8**) was higher than that of radiogallium-labeled unbranched fatty acid ([67Ga]**5** and [67Ga]**7**). The lipophilicity of HBED-CC-conjugated long-chain fatty acids ([67Ga]**5** and [67Ga]**6**) was higher than that of DOTA-Bn-SCN-conjugated corresponding long-chain fatty acids ([67Ga]**7** and [67Ga]**8**). Two benzyl groups of HBED-CC could contribute to this result.

## In vitro stability assay

After the incubation of radiotracers in phosphate buffered saline (PBS) for 24 h at 37°C, more than 90% of all radiogallium-labeled fatty acid derivatives, [67Ga]**5**, [67Ga]**6**, [67Ga]**7**, and [67Ga]**8**, remained intact (Table 2). Murine plasma stabilities in were also not low; however, they were not as high as in PBS.

Rapid imaging protocol for myocardial imaging agent acquired over four to five-minute period using standard camera and over 10-minute period for SPECT camera. An entire stress/rest procedure can be completed in 1 hour. Therefore, in vitro analysis in murine plasma for 1 hour is considered sufficient in this study.

## In vitro protein-binding assays

Bound percentages of [67Ga]**5**, [67Ga]**6**, [67Ga]**7**, and [67Ga]**8** to human serum albumin (HSA) were 99.5 ± 0.0%, 99.9 ± 0.0%, 99.9 ± 0.0%, and 99.9 ± 0.0%, respectively. These results

**Table 2. In vitro stability for radiogallium fatty acid complexes.**

| Radiotracers | In vitro stability | |
|---|---|---|
| | In PBS pH 7.4 (24 h) | In murine plasma (1 h) |
| [$^{67}$Ga]Ga-HBED-CC-PDA ([$^{67}$Ga]**5**) | 96.4 ± 0.5% | 85.0 ± 1.0% |
| [$^{67}$Ga]Ga-HBED-CC-MHDA ([$^{67}$Ga]**6**) | 91.5 ± 1.1% | 88.8 ± 1.4% |
| [$^{67}$Ga]Ga-DOTA-PDA ([$^{67}$Ga]**7**) | 91.1 ± 0.6% | 83.2 ± 1.2% |
| [$^{67}$Ga]Ga-DOTA-MHDA ([$^{67}$Ga]**8**) | 93.3 ± 0.5% | 85.3 ± 1.1% |

Expressed as percentage of remained intact of radiotracer. Data were presented as the mean (SD) for three samples.

indicated that all radiogallium fatty acid complexes have extensive binding ability to HAS-like unmodified fatty acids. HSA is the primary binding protein for fatty acids in the blood to transport them to other tissues [32,33].

## Biodistribution studies

Results of biodistribution of [$^{67}$Ga]**5**, [$^{67}$Ga]**6**, [$^{67}$Ga]**7**, [$^{67}$Ga]**8**, and [$^{123}$I]BMIPP in normal mice are listed in Tables 3–5. The initial myocardial uptakes of four radiogallium-labeled tracers were similar, 5.01 ± 0.30%ID/g, 5.74 ± 1.02%ID/g, 5.67 ± 0.22%ID/g, and 5.29 ± 0.10%ID/g for [$^{67}$Ga]**5** [$^{67}$Ga]**6** [$^{67}$Ga]**7**, and [$^{67}$Ga]**8**, respectively, at 0.5 min postinjection. These uptakes were much lower than that of [$^{123}$I]BMIPP (21.36 ± 2.73%ID/g) and that of even technetium-labeled fatty acid, [$^{99m}$Tc]Tc-MAMA-conjugated hexadecenoic acid (11.22 ± 0.25%ID/g), in a previous study [34]. Long-chain fatty acids are incorporated into myocardial cells through passive diffusion and mainly protein-mediated mechanisms, such as fatty acid transport protein and fatty acid translocase/CD36 [35,36]. The lower myocardial uptake of [$^{67}$Ga]**5**, [$^{67}$Ga]**6**, [$^{67}$Ga]**7**, and [$^{67}$Ga]**8** would be attributed to their lower affinity for fatty acid transporters due to steric hindrance of the Ga-complex site to transporters and a negative charge caused by a free carboxyl group in HBED-CC and DOTA chelates.

On the other hand, it was reported that radiogallium labeled fatty acid analogs, [$^{68}$Ga]Ga-NOTA conjugated 11C fatty acid ([$^{68}$Ga]Ga-NOTA-FA$_{11}$), [$^{68}$Ga]Ga-NOTA conjugated 12C fatty acid ([$^{68}$Ga]Ga-NOTA-FA$_{12}$), [$^{68}$Ga]Ga-NODAGA conjugated 11C fatty acid ([$^{68}$Ga]Ga-NODAGA-FA$_{11}$), [$^{68}$Ga]Ga-DTPA conjugated 11C fatty acid ([$^{68}$Ga]Ga-DTPA-FA$_{11}$), and [$^{68}$Ga]Ga-NOTA conjugated 16C fatty acid ([$^{68}$Ga]Ga-NOTA-FA$_{16}$) (Fig 4) showed the initial heart uptakes of 7.4%ID/gram, 6.4%ID/gram, 3.8%ID/gram, 1.3%ID/gram, and 3.7% ID/gram, respectively, in previous studies [12–14]. In the case of [$^{68}$Ga]Ga-NOTA-FA$_{11}$ and [$^{68}$Ga]Ga-NOTA-FA$_{12}$, although their initial heart uptakes seem to be higher than those of [$^{67}$Ga]**5**, [$^{67}$Ga]**6**, [$^{67}$Ga]**7**, and [$^{67}$Ga]**8**, [$^{68}$Ga]Ga-NOTA-FA$_{11}$ and [$^{68}$Ga]Ga-NOTA-FA$_{12}$ are also not enough for myocardial metabolic imaging because they showed the high radioactivity in the blood and liver and fast clearance from the heart.

An introduction of a β-methyl group in fatty acid analogs, such as [$^{123}$I]BMIPP, has been known to delay the washout from the heart due to interference with its β-oxidation [37]. Thus, we synthesized radiogallium-labeled fatty acid analogs with and without β-methyl group and compared their biodistribution. However, tracer retention in the heart was not greatly changed against our expectation. This may be caused by their low initial uptakes.

High radioactivity in the blood was observed immediately after an i.v. injection of all tracers. Relatively rapid clearance from the blood was observed for [$^{67}$Ga]**5** and [$^{67}$Ga]**6** with an acyclic chelator (HBED), whereas [$^{67}$Ga]**7** and [$^{67}$Ga]**8** with a macrocyclic chelator (DOTA) showed slower blood clearance. The difference of structures between [$^{67}$Ga]**5** and [$^{67}$Ga]**7** and

**Table 3. Biodistribution of [$^{67}$Ga]Ga-HBED-CC-PDA ([$^{67}$Ga]5) and [$^{67}$Ga]Ga-HBED-CC-MHDA ([$^{67}$Ga]6) at 0.5, 2, 5, 20, and 60 min after i.v. injection in ddY mice.**

| Tissues | Time after injection | | | | |
|---|---|---|---|---|---|
| | 0.5 min | 2 min | 5 min | 20 min | 60 min |
| **[$^{67}$Ga]5** | | | | | |
| Blood | 29.91 (2.03) | 17.33 (1.44) | 7.60 (1.04) | 1.10 (0.22) | 0.61 (0.23) |
| Liver | 12.92 (2.48) | 27.82 (1.66) | 37.27 (1.10) | 19.01 (2.74) | 6.17 (0.99) |
| Kidney | 5.14 (0.40) | 3.37 (0.49) | 1.71 (0.16) | 1.00 (0.17) | 1.20 (0.88) |
| Small intestine | 0.98 (0.19) | 1.16 (0.13) | 2.44 (0.23) | 23.09 (2.40) | 26.40 (3.69) |
| Large intestine | 0.34 (0.02) | 0.48 (0.02) | 0.39 (0.01) | 0.23 (0.03) | 0.17 (0.01) |
| Spleen | 1.80 (0.18) | 1.81 (0.26) | 1.15 (0.20) | 0.35 (0.14) | 0.15 (0.03) |
| Pancreas | 1.57 (0.28) | 1.94 (0.28) | 1.48 (0.16) | 0.55 (0.13) | 0.31 (0.09) |
| Lung | 18.70 (3.31) | 6.31 (1.92) | 3.97 (0.56) | 0.94 (0.09) | 0.54 (0.20) |
| Heart | 5.01 (0.30) | 2.89 (0.53) | 1.75 (0.34) | 0.63 (0.06) | 0.25 (0.08) |
| Stomach[†] | 0.41 (0.03) | 0.50 (0.04) | 0.69 (0.38) | 0.60 (0.10) | 0.40 (0.02) |
| Bone | 2.13 (0.46) | 1.91 (0.19) | 1.39 (0.04) | 0.82 (0.12) | 0.81 (0.10) |
| Muscle | 0.79 (0.09) | 0.59 (0.14) | 0.33 (0.08) | 0.30 (0.01) | 0.23 (0.02) |
| Brain | 0.70 (0.07) | 2.46 (0.15) | 0.23 (0.03) | 0.04 (0.01) | 0.02 (0.00) |
| **[$^{67}$Ga]6** | | | | | |
| Blood | 36.78 (3.90) | 27.01 (3.28) | 13.03 (0.12) | 2.53 (0.73) | 0.59 (0.09) |
| Liver | 8.22 (0.86) | 16.39 (1.07) | 20.60 (0.66) | 26.03 (2.46) | 6.80 (0.62) |
| Kidney | 5.57 (0.21) | 4.53 (0.46) | 1.95 (0.22) | 0.88 (0.05) | 0.38 (0.12) |
| Small intestine | 0.63 (0.07) | 0.77 (0.10) | 0.74 (0.22) | 6.26 (2.01) | 23.06 (3.22) |
| Large intestine | 0.33 (0.05) | 0.36 (0.07) | 0.39 (0.05) | 0.23 (0.03) | 0.14 (0.01) |
| Spleen | 1.82 (0.11) | 2.78 (0.08) | 1.84 (0.20) | 0.35 (0.07) | 0.11 (0.00) |
| Pancreas | 1.05 (0.08) | 1.27 (0.09) | 0.97 (0.08) | 0.54 (0.06) | 0.23 (0.06) |
| Lung | 15.31 (3.39) | 12.23 (2.54) | 4.68 (0.32) | 1.25 (0.27) | 0.44 (0.03) |
| Heart | 5.74 (1.02) | 4.51 (0.23) | 2.57 (0.19) | 0.86 (0.12) | 0.33 (0.04) |
| Stomach[†] | 0.32 (0.04) | 0.39 (0.04) | 0.29 (0.15) | 0.61 (0.20) | 1.41 (0.77) |
| Bone | 2.60 (0.07) | 2.40 (0.27) | 1.33 (0.29) | 0.51 (0.05) | 0.30 (0.08) |
| Muscle | 0.94 (0.19) | 0.67 (0.24) | 0.50 (0.09) | 0.20 (0.01) | 0.15 (0.05) |
| Brain | 0.96 (0.01) | 0.81 (0.10) | 0.41 (0.12) | 0.06 (0.01) | 0.01 (0.00) |

Data were presented as %injected dose/gram tissue. Each value represents mean (SD) for three mice.

[†] presented as %ID/organ.

between [$^{67}$Ga]6 and [$^{67}$Ga]8 is only the chelating site. Previously, Jain *et al*. reported that [$^{68}$Ga]Ga-NODAGA-FA$_{11}$ and [$^{68}$Ga]Ga-NOTA-FA$_{11}$, which are $^{68}$Ga-labeled fatty acids with macrocyclic chelators, prolonged the blood clearance compared to [$^{68}$Ga]Ga-DTPA-FA$_{11}$, a $^{68}$Ga-labeled fatty acid with an acyclic chelator (12). Results of this study and a previous study indicate that the kind of chelator used for radiogallium-labeled fatty acids greatly affects the blood clearance of the tracers. Meanwhile, the radioactivity of [$^{67}$Ga]7 and [$^{67}$Ga]8 in the heart was retained, compared to that of [$^{67}$Ga]5 and [$^{67}$Ga]6. The retention in the heart should be caused by the delayed blood clearance.

The high stability of four $^{67}$Ga-labeled fatty acids *in vitro* as mentioned above was reflected in the biodistribution. High accumulation in the bone and delayed blood clearance can be an index of the Ga-complex decomposition in biodistribution studies [38]. Low radioactivity levels in the bone indicate that the Ga-complex decomposition to free Ga metal hardly occurred in four radiotracers.

**Table 4. Biodistribution of [67Ga]Ga-DOTA-PDA ([67Ga]7) and [67Ga]Ga-DOTA-MHDA ([67Ga]8) at 0.5, 2, 5, 20, and 60 min after i.v. injection in ddY mice.**

| Tissues | Time after injection | | | | |
|---|---|---|---|---|---|
| | 0.5 min | 2 min | 5 min | 20 min | 60 min |
| [67Ga]7 | | | | | |
| Blood | 32.14 (1.41) | 28.60 (1.65) | 26.36 (2.41) | 16.43 (0.55) | 10.30 (1.03) |
| Liver | 4.52 (0.20) | 4.75 (0.17) | 4.88 (0.62) | 5.23 (0.56) | 6.50 (0.89) |
| Kidney | 5.19 (0.51) | 4.88 (0.45) | 6.18 (0.20) | 4.31 (0.47) | 4.74 (0.53) |
| Small intestine | 1.15 (0.07) | 1.67 (0.15) | 1.90 (0.26) | 1.64 (0.25) | 1.94 (0.30) |
| Large intestine | 0.54 (0.09) | 0.87 (0.15) | 1.14 (0.18) | 1.14 (0.14) | 0.90 (0.05) |
| Spleen | 1.81 (0.26) | 3.03 (0.45) | 4.29 (0.15) | 2.67 (0.23) | 1.67 (0.09) |
| Pancreas | 2.14 (0.37) | 3.11 (0.54) | 3.70 (0.34) | 2.34 (0.16) | 1.71 (0.26) |
| Lung | 20.57 (1.02) | 14.9 (2.10) | 16.48 (3.59) | 10.89 (1.14) | 7.03 (0.87) |
| Heart | 5.67 (0.22) | 4.76 (0.85) | 6.12 (0.39) | 5.03 (0.36) | 3.28 (0.19) |
| Stomach[†] | 0.55 (0.06) | 0.62 (0.08) | 0.67 (0.06) | 0.61 (0.04) | 0.71 (0.06) |
| Bone | 2.33 (0.54) | 3.07 (0.38) | 4.14 (0.42) | 3.18 (0.43) | 0.23 (0.26) |
| Muscle | 0.66 (0.12) | 0.79 (0.13) | 1.56 (0.17) | 1.63 (0.29) | 1.29 (0.08) |
| Brain | 1.05 (0.15) | 0.89 (0.11) | 0.94 (0.03) | 0.69 (0.33) | 0.34 (0.03) |
| [67Ga]8 | | | | | |
| Blood | 37.07 (0.81) | 33.26 (0.33) | 28.38 (10.58) | 21.65 (2.96) | 13.25 (1.18) |
| Liver | 5.02 (0.24) | 5.16 (0.47) | 4.99 (0.37) | 6.74 (0.31) | 7.88 (1.03) |
| Kidney | 5.71 (0.36) | 6.66 (0.99) | 5.35 (0.34) | 5.18 (0.72) | 4.20 (0.23) |
| Small intestine | 0.98 (0.05) | 1.41 (0.04) | 1.57 (0.04) | 1.92 (0.30) | 2.19 (0.29) |
| Large intestine | 0.38 (0.08) | 0.55 (0.10) | 0.72 (0.04) | 0.96 (0.05) | 0.84 (0.09) |
| Spleen | 1.83 (0.33) | 3.43 (0.20) | 3.31 (0.14) | 3.11 (0.41) | 1.74 (0.11) |
| Pancreas | 1.40 (0.08) | 2.21 (0.04) | 2.09 (0.08) | 2.56 (0.34) | 1.85 (0.15) |
| Lung | 20.04 (0.52) | 18.10 (2.99) | 16.54 (1.36) | 11.95 (0.62) | 7.98 (1.41) |
| Heart | 5.28 (0.10) | 5.83 (0.27) | 5.01 (0.30) | 4.91 (0.39) | 3.62 (0.15) |
| Stomach[†] | 0.42 (0.00) | 0.51 (0.05) | 0.82 (0.46) | 0.59 (0.04) | 0.66 (0.04) |
| Bone | 1.96 (0.06) | 3.17 (0.27) | 2.73 (0.31) | 2.74 (0.34) | 2.18 (0.44) |
| Muscle | 0.59 (0.04) | 0.73 (0.23) | 0.81 (0.22) | 1.17 (0.14) | 1.14 (0.10) |
| Brain | 1.03 (0.23) | 0.98 (0.25) | 0.83 (0.03) | 0.70 (0.25) | 0.43 (0.07) |

Data were presented as %injected dose/gram tissue. Each value represents mean (SD) for three mice.

[†] presented as %ID/organ.

## Metabolite analysis

The results of the metabolite analyses in the blood, heart, and liver at 10 min postinjection of each radiotracer were listed in Table 6. From 64% to 96% of radiotracers were intact in blood, liver, and heart. Two metabolic radioactivity peaks were observed in the heart and liver for [67Ga]7 and liver for [67Ga]8 at HPLC analyses (S7 Fig). In other cases, only one metabolite radioactivity peak was observed below 5 min (S7 Fig).

The results showed that the radiogallium fatty acid analogs were metabolized to more polar radiometabolites. It is known that long-chain fatty acids are metabolized to shorter chain residues by β-oxidation in the heart as well as in the liver [39,40]. Therefore, although the more polar radiometabolites were not determined in this study, the metabolite radioactivity peaks may be shorter chain radiometabolites by β-oxidation of the radiogallium fatty acid analogs. Meanwhile, we assumed that the peak below 5 min is not free [67Ga]Ga$^{3+}$ because Ga-complex decomposition to free Ga metal hardly occurred as above-mentioned.

**Table 5. Biodistribution of [123I]BMIPP at 0.5, 2, 5, 20, and 60 min after i.v. injection in ddY mice.**

| Tissues | Time after injection | | | | |
|---|---|---|---|---|---|
| | 0.5 min | 2 min | 5 min | 20 min | 60 min |
| [123I]BMIPP | | | | | |
| Blood | 20.30 (1.55) | 6.01 (0.74) | 6.75 (1.07) | 10.84 (0.20) | 10.38 (2.03) |
| Liver | 12.44 (2.40) | 21.35 (0.89) | 16.33 (4.41) | 6.54 (0.43) | 4.34 (0.61) |
| Kidney | 5.21 (0.42) | 5.27 (0.61) | 5.93 (1.23) | 8.03 (0.09) | 6.24 (0.62) |
| Small intestine | 0.91 (0.15) | 1.33 (0.23) | 1.39 (0.27) | 1.97 (0.11) | 1.88 (0.29) |
| Large intestine | 0.42 (0.07) | 0.68 (0.14) | 0.66 (0.11) | 1.20 (0.04) | 1.20 (0.15) |
| Spleen | 4.91 (1.27) | 3.76 (1.18) | 3.09 (1.26) | 1.73 (0.55) | 1.68 (0.81) |
| Pancreas | 2.43 (0.68) | 6.11 (1.90) | 5.66 (4.30) | 9.80 (3.25) | 8.04 (3.89) |
| Lung | 24.09 (4.19) | 15.88 (1.94) | 16.87 (3.08) | 16.21 (1.50) | 13.11 (3.05) |
| Heart | 21.36 (2.73) | 27.60 (3.15) | 28.03 (5.37) | 23.41 (3.39) | 20.92 (3.67) |
| Stomach[†] | 0.60 (0.11) | 0.70 (0.17) | 0.95 (0.28) | 1.06 (0.05) | 1.06 (0.17) |
| Bone | 2.63 (0.33) | 1.97 (0.03) | 2.29 (0.27) | 2.66 (0.22) | 2.42 (0.33) |
| Muscle | 2.85 (0.69) | 3.24 (0.49) | 3.42 (1.06) | 3.51 (0.21) | 3.87 (0.80) |
| Brain | 0.51 (0.11) | 0.22 (0.03) | 0.25 (0.03) | 0.43 (0.07) | 0.39 (0.04) |

Data were presented as %injected dose/gram tissue. Each value represents mean (SD) for three mice.

[†] presented as %ID/organ.

## Conclusion

In this study, four novel radiogallium-labeled unbranched and methyl-branched fatty acids ([67Ga]5, [67Ga]6, [67Ga]7, and [67Ga]8) were synthesized and evaluated to assess their feasibility as myocardial metabolic imaging agents. Contrary to our expectation, they cannot be applied as PET myocardial metabolic imaging agents due to their low accumulation in the heart. Therefore, structural modifications, especially in the chelate site, are required to develop $^{68}$Ga-labeled fatty acids with adequate accumulation in the heart as myocardial metabolic imaging agents.

**Fig 4. Chemical structures of [67Ga]Ga-NOTA-FA11, [67Ga]Ga-NOTA-FA12, [67Ga]Ga-NODAGA-FA11, [67Ga]Ga-NOTA-FA16, and [67Ga]Ga-DTPA-FA11.**

**Table 6. Metabolite analyses in the blood, heart, and liver at 10 min postinjection of [67Ga]5, [67Ga]6, [67Ga7, and [67Ga]8 into ddY mice.**

| Tissues | Radiotracers | | | |
|---|---|---|---|---|
| | [$^{67}$Ga]5 | [$^{67}$Ga]6 | [$^{67}$Ga]7 | [$^{67}$Ga]8 |
| Blood | 82.18 (4.82) | 96.14 (0.76) | 70.31 (3.10) | 63.87 (0.70) |
| Heart | 84.17 (5.76) | 83.60 (2.70) | 79.63 (1.23) | 66.64 (2.21) |
| Liver | 95.93 (2.68) | 76.92 (2.95) | 88.38 (2.21) | 77.96 (1.47) |

Data were presented as % of an intact form of [$^{67}$Ga]**5**, [$^{67}$Ga]**6**, [$^{67}$Ga]**7**, or [$^{67}$Ga]**8** determined by TLC. Each value represents mean (SD) for three samples.

## Supporting information

**S1 Fig. HPLC chromatograms of precursors, 1, 2, 3, and 4.**
(PDF)

**S2 Fig. HPLC chromatograms of 5 and [$^{67}$Ga]5.**
(PDF)

**S3 Fig. HPLC chromatograms of 6 and [$^{67}$Ga]6.**
(PDF)

**S4 Fig. HPLC chromatograms of 7 and [$^{67}$Ga]7.**
(PDF)

**S5 Fig. HPLC chromatograms of 8 and [$^{67}$Ga]8.**
(PDF)

**S6 Fig. $^{1}$H-NMR spectra of intermediate compounds.**
(PDF)

**S7 Fig. HPLC chromatograms of metabolite analyses of [$^{67}$Ga]5, [$^{67}$Ga]6, [$^{67}$Ga]7, and [$^{67}$Ga]8.**
(PDF)

**S1 File. Synthesis of 15-aminopentadecanoic acid (APDA).**
(DOCX)

**S2 File. Synthesis of 16-amino-3-methylhexadecanoic acid (AMHDA).**
(DOCX)

**S3 File. Synthesis of HBED-CC(*t*Bu)$_3$-NHS ester.**
(DOCX)

## Acknowledgments

We acknowledge Nur Izni Binti Ramzi for help in manuscript preparation.

## Author Contributions

**Conceptualization:** Kazuma Ogawa.

**Data curation:** Renata Jastrząb.

**Formal analysis:** Nurmaya Effendi, Kenji Mishiro.

**Funding acquisition:** Kazuma Ogawa.

**Investigation:** Nurmaya Effendi, Malwina Gabryel-Skrodzka, Kazuma Ogawa.

**Methodology:** Kenji Mishiro, Junichi Taki, Kazuma Ogawa.

**Project administration:** Kazuma Ogawa.

**Resources:** Hiroshi Wakabayashi, Kazuhiro Shiba, Seigo Kinuya.

**Supervision:** Kenji Mishiro, Kazuma Ogawa.

**Validation:** Nurmaya Effendi.

**Writing – original draft:** Nurmaya Effendi.

**Writing – review & editing:** Kenji Mishiro, Kazuma Ogawa.

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
