## [Decision Letter · Decision Letter 0]

7 Sep 2021

PONE-D-21-24937Synthesis and evaluation of radiogallium-labeled long-chain fatty acid derivatives as myocardial metabolic imaging agentsPLOS ONE

Dear Authors of the manuscript,

Thank you for submitting your manuscript to PLOS ONE. After careful consideration, we feel that it has merit but does not fully meet PLOS ONE’s publication criteria as it currently stands. Therefore, we invite you to submit a revised version of the manuscript that addresses the points raised during the review process.

It is important to show full characterizations and QC of a new probe which is going to studied in animal or human studies.

We look forward to receiving your revised manuscript.

Kind regards,

Anjani Kumar Tiwari, Ph.D.

Academic Editor

PLOS ONE

Journal Requirements:

 [This work was supported in part by Suzuken Memorial Foundation and Kanazawa University SAKIGAKE project 2020.]

Additional Editor Comments:

Dear Author,

The work presented in the manuscript "" Synthesis and evaluation of radiogallium-labeled long-chain fatty acid derivatives as myocardial metabolic imaging agents"

is impressive and can be considered in PLOS One subject to revision suggested by editor.

The author must see the characterization part which seems week as pointed by one reviewer.

Reviewers' comments:

Reviewer's Responses to Questions

**Comments to the Author**

1. Is the manuscript technically sound, and do the data support the conclusions?

Reviewer #1: No

Reviewer #2: Yes

2. Has the statistical analysis been performed appropriately and rigorously? 

Reviewer #1: No

Reviewer #2: Yes

3. Have the authors made all data underlying the findings in their manuscript fully available?

Reviewer #1: Yes

Reviewer #2: Yes

4. Is the manuscript presented in an intelligible fashion and written in standard English?

Reviewer #1: Yes

Reviewer #2: Yes

5. Review Comments to the Author

Reviewer #1: PONE-D-21-24937

Synthesis and evaluation of radiogallium-labeled long-chain fatty acid derivatives as myocardial metabolic imaging agents

1. The scheme shown for the synthesis of Synthesis of 16-Amino-3-methylhexadecanoic acid (AMHDA) is incorrect. The conversion of Dibromo derivative to azide and OH groups is not possible.

2. The mass spectra are missing.

3. The 1H NMR suggest that the compounds are not pure. 13C NMR would provide clear picture, hence are required. The final application of the developed systems is in-vivo, so high purity of the systems is a necessity.

4. The most of the compounds prepared are in mg. Was the quantity sufficient enough to perform the characterization and other studies?

5. What was the specific activity of the Ga derivatives.

6. The stability of the Ga derivatives are in the range of 85% post 1h, which is poor. The metabolite studies are necessary to negate the false positive results.

7. The developed compound even post methylation did not display high uptake in heart thus proves the hypothesis wrong.

8. Also the uptake of the four derivatives in the heart is far less than 123I- BMIPP. Comparative studies with the other known systems would be useful to understand the efficacy of the systems developed.

Reviewer #2: excellent work and good effort to introduce the new chemical scaffold for exploring this area.. this new class of the analogs will further explores to enhance the knowledge of these kind of radio pharmaceuticals

6. PLOS authors have the option to publish the peer review history of their article (what does this mean?). If published, this will include your full peer review and any attached files.

Reviewer #1: No

Reviewer #2: No

---

## [Author Response · Author response to Decision Letter 0]

30 Sep 2021

Reviewer#1: PONE-D-21-24937

We would like to thank the reviewer for the helpful comments. Please read the responses to your comments/suggestions below:

1. The scheme shown for the synthesis of synthesis of 16-Amino-3-methylhexadecanoic acid (AMHDA) is incorrect. The conversion of Dibromo derivative to azide and OH groups is not possible.

Answer: As the reviewer suggested, the structure in the scheme had been incorrect. We corrected the structure in the scheme. The synthetic procedure and 1H NMR data are correct and do not need to be changed.

2. The mass spectra are missing.

Answer: We tried to analyze the molecular weight of 17 using ESI-MS. However, it was difficult. The possible reason could be that this intermediate compound is poorly ionized; therefore, we just showed H-NMR data.

3. The 1H NMR suggest that the compounds are not pure. 13C NMR would provide clear picture, hence are required. The final application of the developed systems is in-vivo, so high purity of the systems is a necessity.

Answer: 1H NMR spectra of these compounds contained broad signals derived from a cyclene ring under equilibrium of some conformers. We tried two different solvents (DMSO d6 and methanol d4), however, the broadening was not solved. 13C NMR also showed broad signal and was difficult to assign. Nevertheless, We think purity of HBED-CC-PDA (1), HBED-CC-MHDA (2), DOTA-PDA (3), and DOTA-MHDA (4) are high because all of non-broad signals of these spectra are derived from these compounds. Moreover, we determined the high purities of 1-4 by HPLC. We showed their chromatograms in Fig S4. 

4. The most of the compounds prepared are in mg. Was the quantity sufficient enough to perform the characterization and other studies?

Answer: 2-3 mg of precursors for the characterization using H-NMR and MS and 100 µg of nonradioactive Ga-complex for the characterization using MS are enough. For other studies using radiolabeled compounds, 20 µg of precursors are needed for radiolabeling. Thus, the quantity we synthesized is sufficient to perform the characterization and other studies.

5. What was the specific activity of the Ga derivatives.

Answer: 

As the reviewer suggested, we added following sentences in Page 19 – 20 line 380 - 388.

“The radiolabeling was performed with carrier-free radionuclides and under non-carrier-added conditions. [67Ga]5 and [67Ga]6 were entirely separated from the corresponding precursors, 1 and 2, by the HPLC purification. Thus, the molar activity of [67Ga]5 and [67Ga]6 was estimated to be approximately 1.5 × 1018 Bq/mol. Meanwhile, the molar activity of [67Ga]7 and [67Ga]8 was determined to be 1.1 × 1015 Bq/mol and 1.2 × 1015 Bq/mol, respectively. These values were much lower than that of [67Ga]5 and [67Ga]6 because [67Ga]7 and [67Ga]8 were not entirely separated from the corresponding precursors.” 

6. The stability of the Ga derivatives are in the range of 85% post 1h, which is poor. The metabolite studies are necessary to negate the false positive results.

Answer: We performed metabolite studies of these radiolabeled compounds in blood, liver, and heart. We attached the data in the revised manuscript. A protocol of the experiment is described at Page 17 – 18 line 337 – 346 and results and discussion of the metabolite study are described at Page 29 line 531 – 549.

7. The developed compound even post methylation did not display high uptake in heart thus proves the hypothesis wrong.

Answer: As the reviewer pointed out, the results were out of our expectations. We expected that the methylation could increase the accumulation of radioactivity in the heart because it is known that methyl-branched fatty acids can prolong cardiac retention. However, the methylation did not affect it. First of all, higher initial uptake in the heart must be necessary. We had mentioned them in our manuscript as follows.

Page 27 line 504 – 509 (result and discussion) “An introduction of a β-methyl group in fatty acid analogs, such as [123I]BMIPP, has been known to delay the washout from the heart due to interference with its β-oxidation (37). Thus, we synthesized radiogallium-labeled fatty acid analogs with and without β-methyl group and compared their biodistribution. However, tracer retention in the heart was not greatly changed against our expectation. This may be caused by their low initial uptakes.” 

Page 30 line 555 – 559 (conclusion) “Contrary to our expectation, they cannot be applied as PET myocardial metabolic imaging agents due to their low accumulation in the heart. Therefore, structural modifications, especially in the chelate site, are required to develop 68Ga-labeled fatty acids with adequate accumulation in the heart as myocardial metabolic imaging agents.”

8. Also the uptake of the four derivatives in the heart is far less than 123I- BMIPP. Comparative studies with the other known systems would be useful to understand the efficacy of the systems developed.

Answer: We added the comparison between radiogallium labeled fatty acids evaluated in this study and those evaluated in previous studies in the results and discussion section.

 

Reviewer #2: excellent work and good effort to introduce the new chemical scaffold for exploring this area. this new class of the analogs will further explores to enhance the knowledge of these kind of radio pharmaceuticals

Answer: Thank you for your appreciation of our study.

---

## [Decision Letter · Decision Letter 1]

29 Nov 2021

Synthesis and evaluation of radiogallium-labeled long-chain fatty acid derivatives as myocardial metabolic imaging agents

PONE-D-21-24937R1

Dear Dr. Ogawa,

We’re pleased to inform you that your manuscript has been judged scientifically suitable for publication and will be formally accepted for publication once it meets all outstanding technical requirements.

Kind regards,

Anjani Kumar Tiwari, Ph.D.

Academic Editor

PLOS ONE

Additional Editor Comments (optional):

All the queries have been answered.

This manuscript is in proper shape for publication.

Reviewers' comments:

Reviewer's Responses to Questions

**Comments to the Author**

1. If the authors have adequately addressed your comments raised in a previous round of review and you feel that this manuscript is now acceptable for publication, you may indicate that here to bypass the “Comments to the Author” section, enter your conflict of interest statement in the “Confidential to Editor” section, and submit your "Accept" recommendation.

Reviewer #2: All comments have been addressed

Reviewer #3: All comments have been addressed

2. Is the manuscript technically sound, and do the data support the conclusions?

Reviewer #2: Yes

Reviewer #3: Yes

3. Has the statistical analysis been performed appropriately and rigorously? 

Reviewer #2: Yes

Reviewer #3: No

4. Have the authors made all data underlying the findings in their manuscript fully available?

Reviewer #2: Yes

Reviewer #3: Yes

5. Is the manuscript presented in an intelligible fashion and written in standard English?

Reviewer #2: (No Response)

Reviewer #3: Yes

6. Review Comments to the Author

Reviewer #2: excellent work need to be published for larger sharing of the results and more future outcomes with this innovative work

Reviewer #3: All required queries have been answered in satisfactory manner.

It can be consider for publication in PLOS ONE.

7. PLOS authors have the option to publish the peer review history of their article (what does this mean?). If published, this will include your full peer review and any attached files.

Reviewer #2: No

Reviewer #3: **Yes: **Dr. Anjani Kumar Tiwari

---

## [Editor Report · Acceptance letter]

6 Dec 2021

PONE-D-21-24937R1 

Synthesis and evaluation of radiogallium-labeled long-chain fatty acid derivatives as myocardial metabolic imaging agents 

Dear Dr. Ogawa:

I'm pleased to inform you that your manuscript has been deemed suitable for publication in PLOS ONE. Congratulations! Your manuscript is now with our production department. 

Kind regards, 

on behalf of

Dr. Anjani Kumar Tiwari 

Academic Editor

PLOS ONE